# Effects of Various Polishing Techniques on the Surface Characteristics of the Ti-6Al-4V Alloy and on Bacterial Adhesion

**Yu-Ting Jhong** [1] , **Chih-Yeh Chao** [2] **, Wei-Chun Hung** [3] **and Je-Kang Du** [4,5,]*

[1]   Ph.D. Program, School of Dentistry, College of Dental Medicine, Kaohsiung Medical University, Kaohsiung 80708, Taiwan; steelwillpower0130@gmail.com

[2]   Department of Mechanical Engineering, National Pingtung University of Science and Technology, Pingtung 91201, Taiwan; cychao@mail.npust.edu.tw

[3]   Department of Microbiology and Immunology, College of Medicine, Kaohsiung Medical University, Kaohsiung 80708, Taiwan; wchung@kmu.edu.tw

[4]   School of Dentistry, College of Dental Medicine, Kaohsiung Medical University, Kaohsiung 80708, Taiwan

[5]   Department of Dentistry, Kaohsiung Medical University Hospital, Kaohsiung 80708, Taiwan

*   Correspondence: dujekang@gmail.com

**Abstract:** Ti-6Al-4V, although widely used in dental materials, causes peri-implant inflammation due to the long-term accumulation of bacteria around the implant, resulting in bone loss and eventual failure of the implant. This study aims to overcome the problem of dental implant infection by analyzing the influence of Ti-6Al-4V surface characteristics on the quantity of accumulated bacteria. Ti-6Al-4V specimens, each with different surface roughness are produced by mechanical, chemical, and electrolytic polishing. The surface roughness, surface contact angle, surface oxygen content, and surface structure were measured via atomic force microscopy (AFM), laser scanning confocal microscopy (LSCM), drop shape analysis (using sessile drop), X-ray photoelectron spectroscopy (XPS), and X-ray diffraction (XRD). The micro and macro surface roughness are 10.33–120.05 nm and 0.68–2.34 μm, respectively. The surface X direction and Y direction contact angle are 21.38°–96.44° and 18.37°–92.72°, respectively. The surface oxygen content is 47.36–59.89 at.%. The number of colonies and the optical density (OD) are $7.87 \times 10^6$–$17.73 \times 10^6$ CFU/mL and 0.189–0.245, respectively. The bacterial inhibition were the most effective under the electrolytic polishing of Ti-6Al-4V. The electrolytic polishing of Ti-6Al-4V exhibited the best surface characteristics: the surface roughness of 10 nm, surface contact angle of 92°, and surface oxygen content of 54 at.%, respectively. This provides the best surface treatment of Ti-6Al-4V in dental implants.

**Keywords:** Ti-6Al-4V; surface roughness; surface contact angle; surface oxygen content; biofilm; colony

## 1. Introduction

According to possessed well corrosion resistance and biocompatibility, Ti-6Al-4V alloy is widely used in biomedical materials [1] such as, hip joints, intervertebral implants, cardiac catheters, and dental implants [2]. According to clinical statistics on dental applications [2], the success rate of dental implants is approximately 97.3% [3], and the failure rate is approximately 1.9%–41.4% [4]. The main reasons for failure in the order of decreasing frequency are included periodontitis (81.3%), oral hygiene habits (47%–65%), smoking (11.4%–64.3%), systematic disease (8.9%–26%), infection during surgery (18.7%), and radio-therapy (11.5%) [5,6]. It can also be seen that the main cause of the failure is the infection caused by the accumulation of bacteria [7].

The strain of Streptococcus was called "early colonizers" because they take part in formation of the early attachment of biofilm. A strain of Streptococcus can make a lot of extracellular polysaccharides when they gain the sucrose. Then, they can strengthen the mechanical properties and adhesiveness of the biofilm and cause biological complications [8]. The biofilms have influence on inflammatory response and any bone destroy [9]. Therefore, it is necessary to decrease early bacterial attachment to prevent biological complications.

In order to improve the infection of dental implants, using antibiotics or drugs often treats infections of dental implant complications. However, the antibiotic resistance is reaching 71.7% [10]. Therefore, it is necessary to study the inhibition or adhesion characteristics of the metal surface characteristics of the implant. In past research, the optical density (OD) [11] has often been used to measure biofilms. The colony forming unit (CFU) [12] was used to measure the number of colonies to estimate the number of accumulated bacteria. The variation factors for the inhibition or adhesion characteristics of bacteria are mainly the surface characteristics of the implant, including surface roughness [13–16], contact angle [17,18], and oxygen content [19–21].

Surface alloying [22,23], amorphous treatment [24,25], and modification of surface properties [13, 17,19] are the various methods for changing the surface characteristics. The improved version of surface alloying includes spraying metal elements, such as Cu [22] and Ag [23]; the Cu and Ag ions released from the surface of the implants, pass through the bacterial cell wall into the cell membrane of the bacteria, stopping the metabolic growth of the bacteria, killing them, and thereby, achieving antibacterial activity [26,27]. Surface alloying can not only increase the antibacterial properties but can also improve the wear resistance and corrosion resistance of the material. However, there are still doubts about the compatibility of antibacterial properties of metal elements within the human body, and excessive amounts may be toxic to human cells [28]. Amorphous treatment includes the use of organic forms of natural fungicides, such as positively charged chitosan (chitosan) [24], and inorganic forms of amorphous metals (such as iron, chromium, and nickel) [25].

Additionally, the methods to modify the surface properties include mechanical, chemical, and electrolytic polishing [29–32]. Mechanical polishing is mainly used to grind the surface of the material with granular media to reduce the roughness of the surface. This method can only be used on the surface of a workpiece with less complex shapes, and the surface roughness value can only reach approximately 0.3–0.6 μm, and, it is difficult to obtain a mirror surface (approximately 0.13 μm) [29]. Chemical polishing mainly uses chemical liquids to dissolve the oxide layer. Chemical polishing can achieve surface roughness less than 1 μm and is also used for complex shapes surfaces. However, it is difficult to control the polishing condition, and this hinders the attainment of a good polishing effect [31]. Electrolytic polishing can achieve a smooth surface roughness of less than 10 nm, and to improve the surface characteristics antibacterial effects are attained. The disadvantage of the process is that the parameters are difficult to control; therefore, it is difficult to find the exact values of the parameters suitable for the material [32].

In terms of surface characteristics, some researchers have proposed that the accumulation of bacteria is closely related to the interaction between the characteristics of the surface of the material, including roughness [13–16], the surface contact angle [33], and the surface oxygen content [19]. Studies have shown that a surface roughness of less than 0.2 μm can effectively reduce the accumulation of bacteria [14,15]. However, some researchers believe that when the surface roughness is less than 0.2 μm, there is no significant difference in the adhesion of bacteria [16]. Furthermore, the quantity of bacteria attached decreases as the contact angle increases [33]. However, other studies have suggested that bacteria will affect the amount of adhesion according to different surface topography. The adhesion of bacteria is related to the extracellular polymeric substances (EPS) produced by bacteria, and the contact angle has a small effect on the extent of bacterial adhesion [17]. With respect to the surface oxygen content, the surface oxygen elements and the thickness of the oxide layer also affect the adhesion of bacteria. When the surface structure contains oxygen and the thickness of the oxide layer is greater than 1.7–5 nm, bacteria easily accumulate to form biofilms [19]. Another study proposed that the

active oxygen in the oxide layer can decompose the oxide film to achieve the effect of inhibiting the formation of bacteria [20,21].

In summary, the results of previous research on the influence of surface roughness, contact angle, and surface oxygen content on the quantity of accumulated bacteria are not consistent. And Futhermore, some of the previous studies only research among these two factors. There are rare studies that discuss the influence of the aforementioned three factors on the quantity of accumulated bacteria at the same time. In order to understand the degree of influence of the variation in these three surface factors on the quantity of accumulated bacteria, this paper discusses the different surface polishing methods of Ti-6Al-4V ELI (Extra Low Interstitials) to produce different surface characteristics, as well as through statistical analysis of data through one-way Analysis of variance (ANOVA), this paper discusses the degree of influence of the surface roughness, contact angle, and surface oxygen content on bacterial adhesion by *Streptococcus mutans*.

## 2. Materials and Methods

### 2.1. Specimens Preparation

The specimens were obtained from a grade 5 titanium bar (Ti-6Al-4V ELI, ASTM F136, Titanium Industries Inc, Rockaway, NJ, USA) with Ø = 12 mm. The specimens were cut to a thickness of 1 mm using a computer numerical control (CNC) machine (SR 20J Type-C, RDMO Machine-tools, Contamine-sur-Arve, Auvergne-Rhône-Alpes, France) at 1000 rpm. The specimens after cutting by CNC without surface treatment were the control group. The experimental groups are following below the surface treatment steps. For mechanical polishing, 1000# and 1500# silicon carbide papers were used. In addition, for chemical polishing (CP), the specimens were immersed in a chemical solution of 5% hydrofluoric acid, 30% nitric acid, and 65% deionized water, at 25 °C for 2 min. Electropolishing was conducted by exposing from the CNC cutting specimens to an area of Ø = 12 mm × 1 mm. Electropolishing was conducted by stirring the electrolyte mixed with 83% acetic acid, 22% perchloric acid, and 5% glycerol for 100, 200, and 300 s, respectively. Additionally, electropolishing was conducted under 25–30 V and approximately 0.5–1 A, and the distance between the anode and the cathode was 30 mm. Moreover, the temperature of the electrolyte was controlled between −15 and −10 using a low temperature circulator (CA-1111, EYELA, San Diego, CA, USA). In order to remove the residual reactants on the surface, the electrolyte was stirred at 300 rpm with a magnetic bar by stirring a hot plate (PC420D, CORNING, New York City, NY, USA). Finally, all the specimens were cleaned using an ultrasonic cleaner (O-LEO-801, Blossom, Kaohsiung, Taiwan) for 15 min with acetone, for 15 min with deionized water, then for 15 min with ethanol (99%), and finally dried for 15 min. Meanwhile, all specimens were kept in a vacuum. The specimens were divided into seven groups with different surface treatments. In Table 1, it shows that experimental specimen marked as A-F and control specimen G. Where, the samples A–C are electropolishing for 300, 200, and 100 s, respectively. Being chemically polished for 2 min is designed as specimen D. Futhermore, sample E and F are mechanically polishing by using of 1000# and 1500# silicon carbide papers. In addition, the control specimen G is obtained by cutting using a CNC machine. Each examinational test would be analyzed 3 samples.

**Table 1.** Group of Ti-6Al-4V during different polishing process.

| Group | Specimen | Polishing Process |
|---|---|---|
| Experimental | A | 300 s electric Polishing |
| | B | 200 s electric Polishing |
| | C | 100 s electric Polishing |
| | D | 2 min Chemical Polishing |
| | E | #1500 SiC |
| | F | #1000 SiC |
| Control | G | After cutting without surface treatment |

## 2.2. Surface Characterization

All the topographical features of the specimens of the present study were identified via scanning electron microscopy (SEM, JEOL JSM-6380, Tokyo, Japan) at 20 keV. Ti-6Al-4V specimens after different polishing treatment were immersed in *S. mutans* suspension. Specimens were fixed in 2.5 vol.% glutaraldehyde at 4 °C for 2 h, and then dehydrated step-wise with series alcohol concentrations (50 vol.%, 60 vol.%, 70 vol.%, 80 vol.%, 90 vol.%, and 100 vol.%) before the SEM observation. Subsequently, in order to improve their conductivity, using gold film was used to cover the specimen surfaces. The biofilm morphologies of Ti-6Al-4V specimens after different polishing treatment were evaluated by using a SEM (JEOL JSM-6380, Tokyo, Japan). The micro surface roughness and topography were analyzed via atomic force electron microscopy (AFM, ARDIC P150, Taipei, Taiwan). The scan area of the surface roughness was 50 μm × 50 μm. In addition, the macro surface roughness was determined via laser scanning confocal microscopy (LSCM, VK-X250, Keyence, Osaka, Japan). The laser wavelength was 405 nm, and images were taken at 20× magnification and analyzed using Zen Blue software (2010, Keyence, Osaka, Japan). The values of surface roughness were expressed as arithmetic mean deviation (*R*a), and all the groups were measured by three specimens.

Contact angle measurements were conducted using an FTA 1000 drop shape analysis system (First Ten Angstroms, Portsmouth, VA, USA) under 20 °C ambient conditions. For the sessile drop method, the present study used deionized water with a volume of 1 μL, which was provided with a micro syringe. The contact angle results were analyzed by recording 100 pieces of photos using the software FTA22 (Drop snake analysis, Portsmouth, VA, USA). The direction of the acquired image of the droplet profile on the different polished surface regarding to the polish direction was as shown as Figure 1. The X direction is perpendicular, and the Y direction is parallel. The contact angle data were tested three times for obtaining the average and SD values.

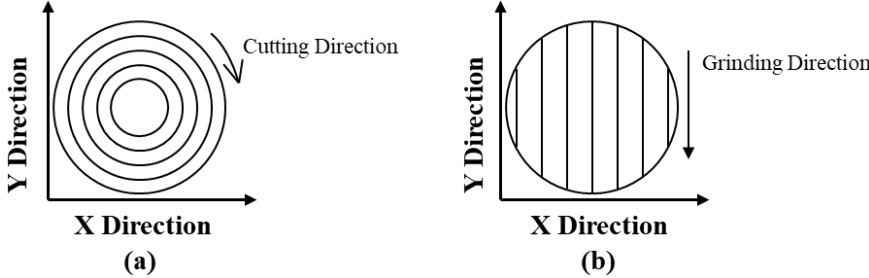

**Figure 1.** The contact angle measurement direction of the acquired image of the droplet profile on the different polished surface. The X direction is perpendicular, and the Y direction is parallel. (**a**) The texture is Ti-6Al-4V after CNC machine cutting and chemical polishing. (**b**) The texture is Ti-6Al-4V after mechanical polishing.

In addition, the present study conducted phase analysis via X-ray diffraction (XRD). An X-ray diffractometer (D8 Advance, Bruker AXS GmbH, Karlsruhe, Germany) with a current of 40 mA and a voltage of 40 mV was used with Cu K$\alpha$ radiation and run with a step size 2$\theta$ of 0.4.

X-ray photoelectron spectroscopy (XPS, JEOL, JAMP-9500F, Peabody, MA, USA) was used to analyze the surface elements and electronic states and recorded using an Al K$\alpha$ source at 150 W. The XPS analysis area was 8 mm × 9 mm. The binding energy (BE) scale refers to the C1*s* and is calibrated at 285.0 eV.

### 2.3. Biological Analysis

Bacterial strains and growth conditions were conducted for this study. *Streptococcus mutans* (*S. mutans*) ATCC 25175 was grown under microaerophilic conditions for 24 h at 37 °C on Brain Heart Infusion (BHI) agar plates (Sigma-Aldrich, St. Louis, MI, USA), supplemented with 3 g/L of yeast extract (Sigma-Aldrich, St. Louis, MI, USA) and 200 g/L of sucrose (Sigma, Winston, Oakville, ON, Canada). After incubation, approximately $10^6$ colony-forming units (CFUs) of *S. mutans* cells were inoculated in BHI broth with 200 g/L of sucrose. All the specimens were placed into 24 well-plates and immersed in a bacterial suspension containing 500 μL of *S. mutans,* each well was covered for 48 h at 37 °C. Thereafter, the specimens were washed twice with phosphate buffer solution (PBS, Sigma–Aldrich, St. Louis, MI, USA).

Ti-6Al-4V specimens were recovered from the 24 well-plates after incubation in *S. mutans suspension*. All the specimens were transferred to new plates for the evaluation of biomass by the crystal violet (CV) staining method, and the biofilm was measured using a spectrophotometer (Epoch, BioTek®, Vermont, CA, USA) to determine the optical density ($OD_{550\,nm}$) in a microplate reader. Biofilm formation and analysis was conducted through the following steps: (1) The specimen was washed twice in PBS. (2) The washed specimen was placed in a 5 mL eppendorf tube with 1mL PBS and thereafter, the adherent bacterial was collected by vortexing (Genie 2, BERTEC, Taipei, Taiwan) treatment for 1 min and gain bacterial suspension. (3) The bacterial suspension was serially diluted ($10^{-3}$, $10^{-4}$, $10^{-5}$, $10^{-6}$, and $10^{-7}$). (4) The diluted solutions were used for repeated plate smearing and analysis of bacteriostatic effects. (5) The suspension was then diluted (up to $10^{-6}$ dilution) in PBS and plated on BHI agar to quantify CFUs /mL. These experiments were performed three times and conducted in three independent assays.

### 2.4. Statistical Analysis

Statistical analyses were conducted using the statistical software SPSS 20.0 (IBM, Armonk, NY, USA). Statistical analysis of the data indicated values as mean ± standard deviation, as shown in Table 2. In each experiment, triple replicates of each surface were used. The difference in surface roughness results obtained via LSCM and AFM, contact angle, value of OD, and colony of *Streptococcus mutans* (CFU/mL) were analyzed by one-way ANOVA, followed by a least significant difference (LSD) test. Statistical differences with $p < 0.05$ were considered to be statistically significant.

**Table 2.** The values of the OD, CFU, roughness, contact angles and oxygen content of the present alloys, specimens A–G.

| Specimen | $OD_{550\,nm}$ ($n = 3$) | *S. Mutans* $10^6$ (CFU/mL) ($n = 3$) | Micro Roughness (nm) ($n = 3$) | Macro Roughness (μm) ($n = 3$) | Contact Angle(°) X Direction ($n = 3$) | Contact Angle (°) Y Direction ($n = 3$) | Oxygen Atomic Percentage (%) ($n = 3$) |
|---|---|---|---|---|---|---|---|
| A | 0.189 ± 0.021 [a] | 7.87 ± 2.23 [a] | 10.33 ± 1.14 [a] | 0.68 ± 0.03 [a] | 96.44 ± 4.84 [a] | 92.72 ± 2.4 [a] | 53.89 ± 0.50 [a] |
| B | 0.204 ± 0.021 [a,b] | 9.00 ± 2.44 [a,b] | 12.63 ± 0.81 [a] | 0.75 ± 0.05 [a] | 84.89 ± 1.72 [b] | 88.61 ± 2.84 [a,b] | 52.35 ± 0.48 [b] |
| C | 0.203 ± 0.023 [a,b,c] | 10.33 ± 1.51 [a,b,c] | 58.72 ± 3.68 [b] | 1.68 ± 0.02 [b] | 85.39 ± 3.17 [b] | 84.53 ± 7.64 [b] | 51.32 ± 0.47 [c] |
| D | 0.217 ± 0.012 [b,c,d] | 10.90±2.29 [b,c,d] | 74.08 ± 9.15 [c] | 1.81±0.23 [c] | 59.01 ± 1.11 [c] | 56.05 ± 2.91 [c] | 50.42 ± 0.33 [d] |
| E | 0.219 ± 0.019 [b,c,d] | 12.35 ± 1.18 [c,d,e] | 86.42 ± 2.05 [d] | 1.82 ± 0.09 [c] | 44.22 ± 0.67 [d] | 37.63 ± 6.09 [d] | 48.44 ± 0.76 [e] |
| F | 0.231 ± 0.018 [d,e] | 14.00 ± 2.90 [e] | 98.30 ± 3.79 [e] | 2.04 ± 0.03 [d] | 29.88 ± 1.73 [e] | 22.14 ± 3.07 [e] | 47.49 ± 0.76 [f] |
| G | 0.245 ± 0.013 [e] | 17.73 ± 2.54 [f] | 120.05 ± 7.89 [e] | 2.34 ± 0.07 [e] | 21.38 ± 1.41 [f] | 18.37 ± 0.61 [e] | 47.36 ± 0.93 [f] |

a, b, c, d, e, f: Within each column, the same superscript letters indicate homogeneous subsets ($p > 0.05$), and different superscript letters indicate statistical significance ($p < 0.05$) following one-way ANOVA and post-hoc tests.

## 3. Results

Results and their statistical significance are presented in Table 2. The polishing process of the specimen is also listed in Table 1 (Specimen G—raw material obtained by CNC cutting, Specimen F—mechanically polished by #1000 SiC, Specimen E—mechanically polished by #1500 SiC, Specimen D—chemically polished for 2 min, Specimen C—electropolished for 100 s, Specimen B—electropolished

for 200 s, and Specimen A—electropolished for 300 s). The macroscopic and microscopic surface roughness of the Ti-6Al-4V alloy with various polishing processes using the LSCM and AFM methods are in the ranges of 10–120 nm and approximately 0.5–2.4 µm, respectively. The analysis of the surface contact angle revealed that the contact angle value was between 15° and 100°; the surface oxygen content was between 47 at.% and 54 at.% after different conducting polishing procedures of Ti-6Al-4V. Ti-6Al-4V after elepolishing have the lowest surface roughness and the highest contact angle and the most oxygen atomic percentage leads to the lowest amount of *Streptococcus mutans* to attaching to the Ti-6Al-4V surface. The present study shows Ti-6Al-4V after elepolishing have the most effect of *Streptococcus mutans* inhibition.

Additionally, the micro and macro surface roughness are shown in Figure 2. It is revealing that micro and macro surface roughness are 10.33–120.05 nm and 0.68–2.34 µm, respectively. The AFM micrographs, LSCM micrographs (the color is representative surface roughness; when the color is different, it is meaning the surface has different height), and SEM micrographs are also shown in Figure 3a–c. The surfaces of specimen A, specimen B, and specimen C are smooth, it is revealing less undulation in AFM morphology; however, macroscopic scratches and undulations are observed on the other surfaces of the specimens. Specimen D is chemical polishing and the SEM image show the directional scratches and α phase base and β phase. AFM micrographs of specimen D shows more undulate than specimen A, B, and C. Specimen E, F, and G show the scratches of consistent direction. AFM and LSCM micrographs are revealing the consistent morphology.

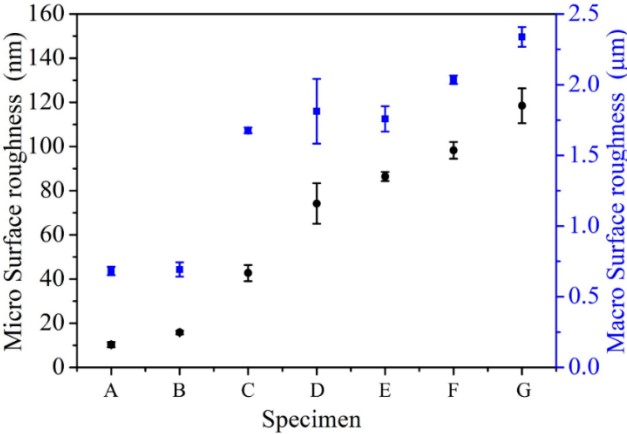

**Figure 2.** Surface roughness of the specimen A–G were evaluated via AFM, and LSCM after various surface treatments: (Specimen A—electropolished for 300 s, Specimen B—electropolished for 200 s, Specimen C—electropolished for 100 s, Specimen D—chemically polished for 2 min, Specimen E—mechanically polished by #1500 SiC, Specimen F—mechanically polished by #1000 SiC and Specimen G-obtained after CNC cutting of pristine material).

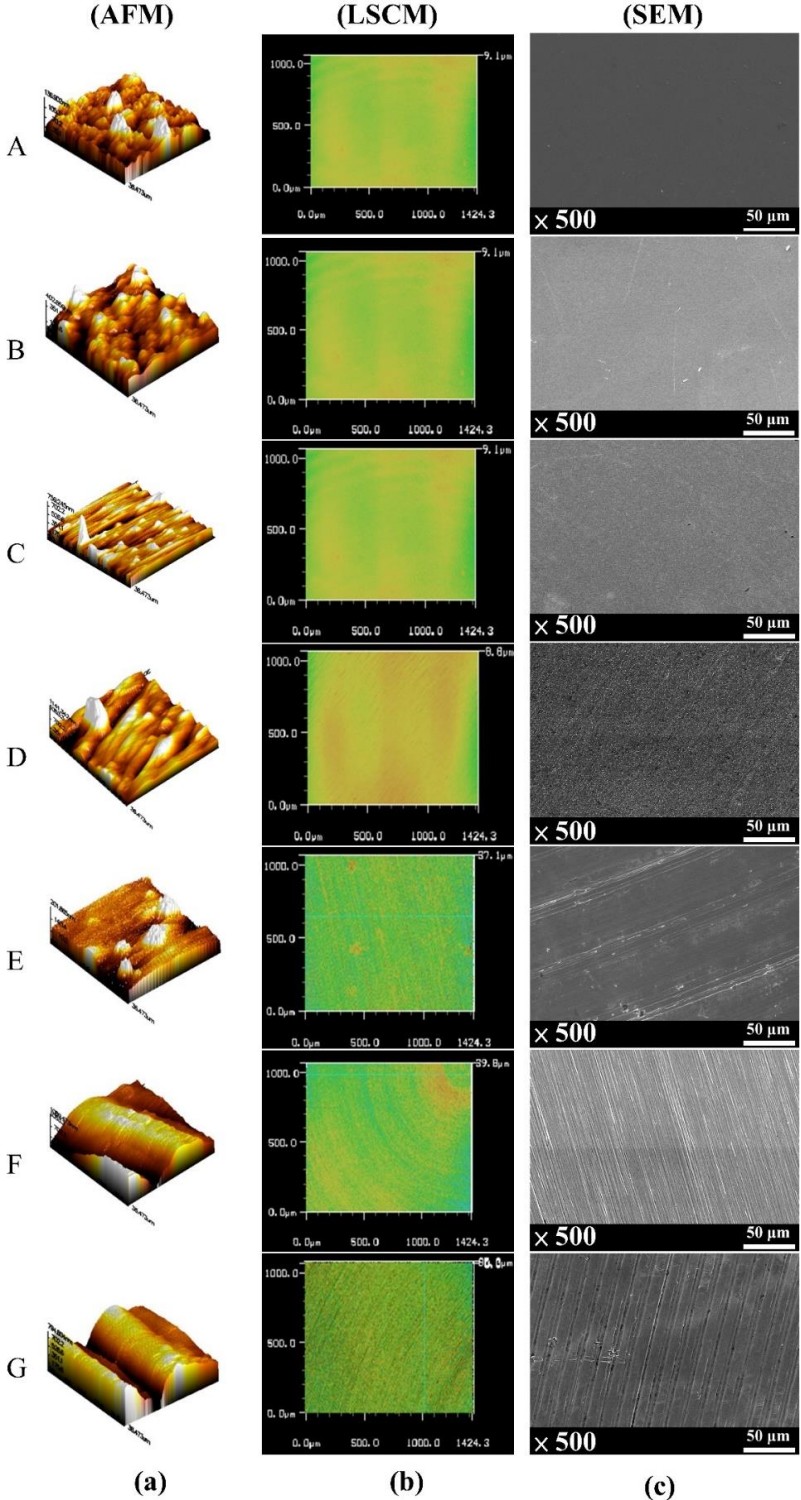

**Figure 3.** (**a**) AFM micrographs, (**b**) LSCM micrographs, and (**c**) SEM micrographs of Specimen A–G. AFM micrographs is showing that surface roughness of specimen A–G are changed from low to high. LSCM micrographs is revealing macro surface roughness of specimen A–G. The SEM image of is indicates specimen A–C show smooth surface and specimen D–G show directional scratches. (Specimen A—electropolished for 300 s, Specimen B—electropolished for 200 s, Specimen C—electropolished for 100 s, Specimen D—chemically polished for 2 min, Specimen E-mechanically polished by #1500 SiC, Specimen F—mechanically polished by #1000 SiC and Specimen G—obtained after CNC cutting of pristine material).

Figure 4 shows the surface contact angles of the test pieces A–G. Among them, the surface contact angles of the test pieces A and B are the largest, while the surface contact angle of test piece G is the smallest. Contact angles of the present specimen A–G with the Figure 4a X direction and Figure 4b Y direction revealing 21.38°–96.44° and 18.37°–92.72°, respectively, and contact angles of X directions more than the Y direction. Figure 5 shows contact angles images of the present specimen A–G with the Figure 5a X direction and Figure 5b Y direction, and they have consistent variety of contact angle. The variant contact angles of a parallel and perpendicular nature are 0.86°–7.74°. Figure 6 shows that the smaller the surface roughness of the alloy, the larger the contact angle. The surface roughness and contact angle shows a linear relationship with a negative correlation. The linear equation of the surface X direction contact angle and microscopic surface roughness is $y = -0.02x + 2.71$ (slope is −0.02). The linear equation of the surface X direction contact angle and macroscopic surface roughness is $y = -1.32x + 145.36$ (the slope is −1.32). The linear equation of the surface Y direction contact angle and microscopic surface roughness is $y = -0.01x + 2.57$ (slope is −0.01). The linear equation of the surface Y direction contact angle and macroscopic surface roughness is $y = -1.22x + 135.76$ (the slope is −1.26). Therefore, surface roughness and contact angle have negative correlation. The surface roughness is decreases when the contact angle increases. It enhances hydrophobicity because of electropolishing, and it is demonstrated that contact angle resulted from the decreased surface roughness.

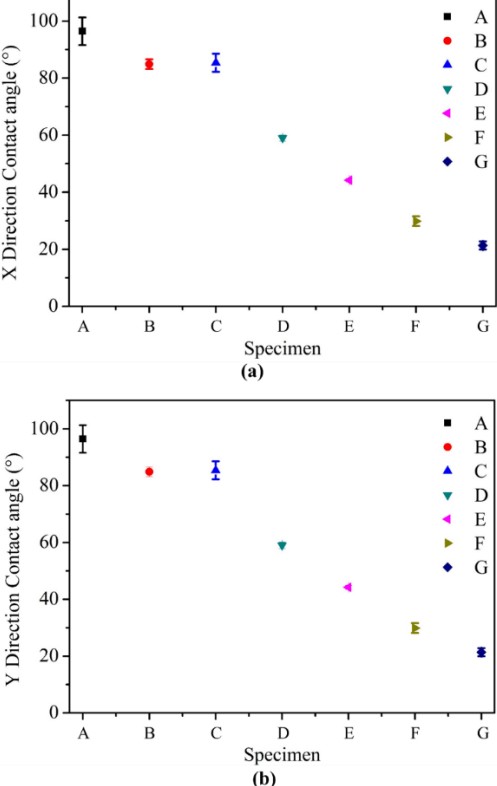

**Figure 4.** Contact angles of the present specimen A–G with (**a**) X direction and (**b**) Y direction revealing 21.38°–96.44° and 18.37°–92.72°, respectively, and contact angles of X directions more than the Y direction. (Specimen A—electropolished for 300 s, Specimen B—electropolished for 200 s, Specimen C—electropolished for 100 s, Specimen D—chemically polished for 2 min, Specimen E—mechanically polished by #1500 SiC, Specimen F—mechanically polished by #1000 SiC, and Specimen G—obtained after CNC cutting of pristine material).

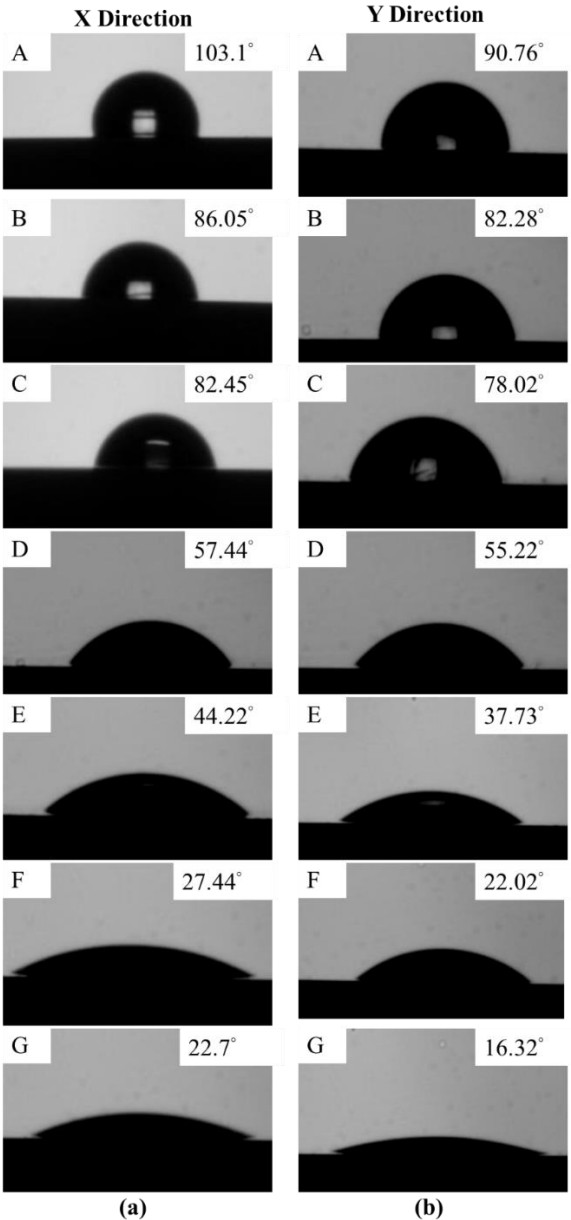

**Figure 5.** Contact angles images of the present specimen A–G with (**a**) X direction and (**b**) Y direction, and they have consistent variety among specimens A–G. (Specimen A—electropolished for 300 s, Specimen B—electropolished for 200 s, Specimen C—electropolished for 100 s, Specimen D—chemically polished for 2 min, Specimen E-mechanically polished by #1500 SiC, Specimen F—mechanically polished by #1000 SiC, and Specimen G-obtained after CNC cutting of pristine material).

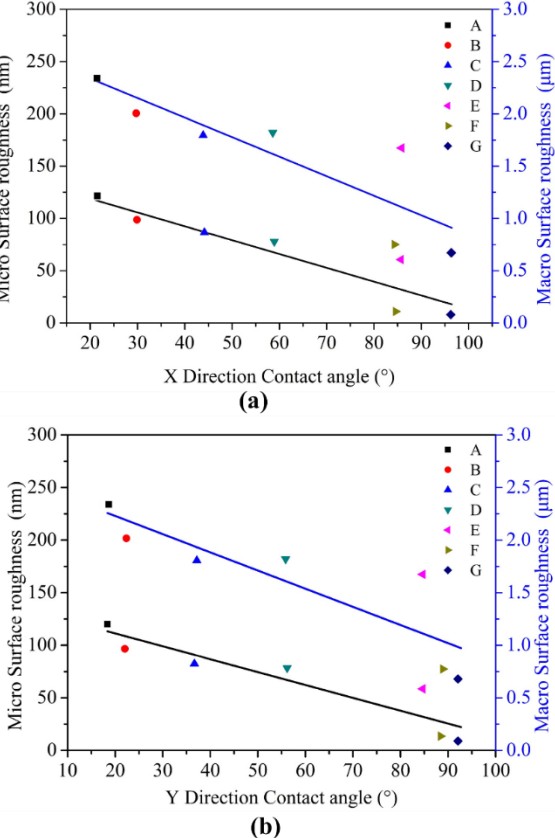

**Figure 6.** Relationship between contact angle of the present specimen A–G and micro and macro roughness with (**a**) X direction and (**b**) Y direction. The linear equation of the surface X direction contact angle and microscopic surface roughness is $y = -0.02x + 2.71$ (slope is $-0.02$). The linear equation of the surface contact angle and macroscopic surface roughness is $y = -1.32x + 145.36$ (the slope is $-1.32$). The linear equation of the surface Y direction contact angle and microscopic surface roughness is $y = -0.01x + 2.57$ (slope is $-0.01$). The linear equation of the surface contact angle and macroscopic surface roughness is $y = -1.22x + 135.76$ (the slope is $-1.22$) (Specimen A—electropolished for 300 s, Specimen B—electropolished for 200 s, Specimen C—electropolished for 100 s, Specimen D—chemically polished for 2 min, Specimen E—mechanically polished by #1500 SiC, Specimen F—mechanically polished by #1000 SiC, and Specimen G—obtained after CNC cutting of pristine material).

Figure 7a is a typical SEM image of Ti-6Al-4V, showing that the basic structure of the alloy comprises an equiaxed $\alpha$ phase base and island $\beta$ phase. Figure 7b shows the XRD data. In addition to the $\alpha$-phase and $\beta$-phase diffraction peaks, the $\alpha TiO_2$ phase diffraction peaks can also be observed. This shows that there is a certain titanium dioxide structure in the present titanium alloy. The equiaxed $\alpha$ phase was $\alpha$-Ti structure with the representative triangle symbol and island $\beta$ phase with the representative circle symbol are shown in Figure 7b.

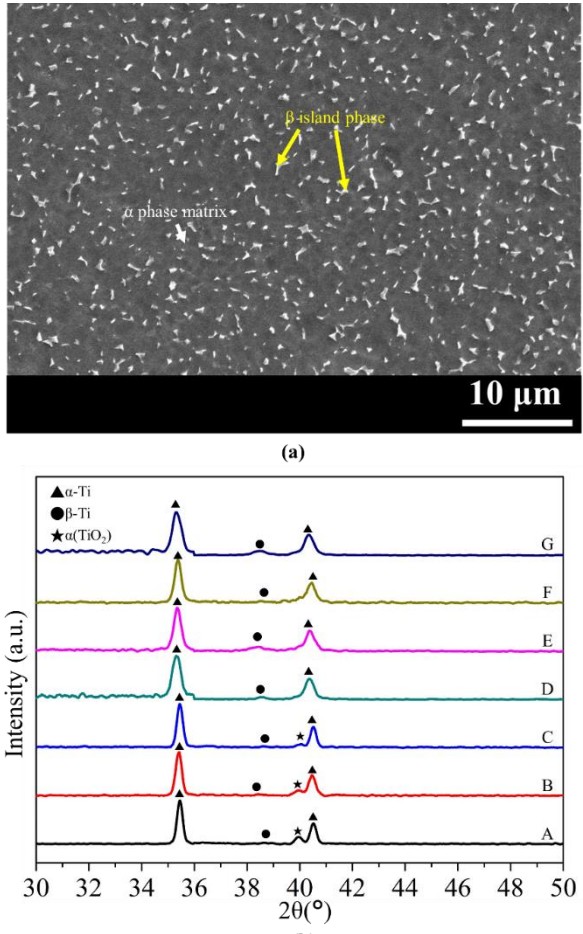

**Figure 7.** (**a**) A typical SEM micrograph of control specimen G, revealing that an equiaxed α-phase matrix and island β-phase particles occurred, (**b**) a XRD profiles of the specimen A–G, revealing that the diffraction peak of α-phase, β-phase and α-TiO$_2$ precipitates occurred (Specimen A—electropolished for 300 s, Specimen B—electropolished for 200 s, Specimen C—electropolished for 100 s, Specimen D—chemically polished for 2 min, Specimen E—mechanically polished by #1500 SiC, Specimen F—mechanically polished by #1000 SiC and Specimen G—obtained after CNC cutting of pristine material).

Figure 8a shows the XPS analysis of Ti-6Al-4V after different polishing treatments; all specimens are showing that there are Al2$p$, C1$s$, Ti2$p$, O1$s$, and V2$p$ elements on the surface. Figure 8b is O1$s$ binding energy in each layer of specimens G and shows the O1$s$ bond energy of Ti-6Al-4V after different polishing treatments through the 532 eV energy analysis diagram (the bond energy is the average value of energy required for each chemical bond when gaseous molecules are disassembled into gaseous atoms under standard conditions, or the atom from which the electrons derive the orbital binding energy). It can be observed that O1$s$ has a higher energy value on the display surface between 0.2 and 1.4 μm in depth.

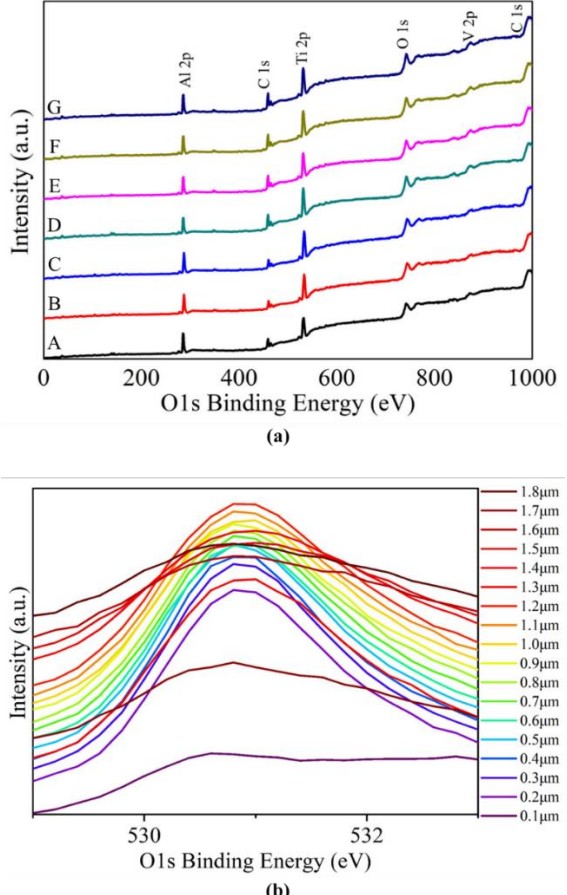

**Figure 8.** (**a**) XPS-spectra of the present specimen A–G, revealing that Al2*p*, C1*s*, Ti2*p*, O1*s*, and V2*p* elements on the surface. (**b**) O1*s* binding energy in each layer of specimen A revealing O1s has a higher energy value on the display surface between 0.2 and 1.4 μm in depth. (Specimen A—electropolished for 300 s, Specimen B—electropolished for 200 s., Specimen C—electropolished for 100 s, Specimen D—chemically polished for 2 min, Specimen E—mechanically polished by #1500 SiC, Specimen F—mechanically polished by #1000 SiC, and Specimen G—obtained after CNC cutting of pristine material).

Figure 9a shows the distribution of the typical O1*s* bond energy around 532 eV after Ti-6Al-4V is scanned at every 0.1 μm in depth after different polishing treatments. At the depth of approximately 0.2–1.4 μm, a high peak value of O1*s* can be observed, indicating that the oxygen content is high in this interval. Figure 9b shows the distribution of O1*s* bond energy and oxygen content (integrated value of the peak near 532 eV from Figure 8b in the depth range of 1–1.8 μm from the surface of the alloy after different polishing procedures. After integrating the wave peak area, the atomic percentage of the relative oxygen content of the alloy after different polishing procedures can be obtained. It can be found that the atomic percentage of oxygen content is relatively high, approximately 45 at.%–60 at.% at a depth of approximately 0.2–1.4 μm. Specimens A, B, and C (electropolishing) have 51.32 at.%–53.89 at.% oxygen content and are showing the highest oxygen content. Specimen D has 50.42 at.% oxygen content, and is lower than electropolishing. Specimens E and F have 47.49 at.% and 48.44 at.% oxygen content and are lower than chemical polishing. Specimen G has a natural oxide layer (47.36 at.% oxygen content) and has the lowest oxygen content. Therefore, the order of oxygen content of polishing process is electropolishing > chemical polishing > mechanical polishing.

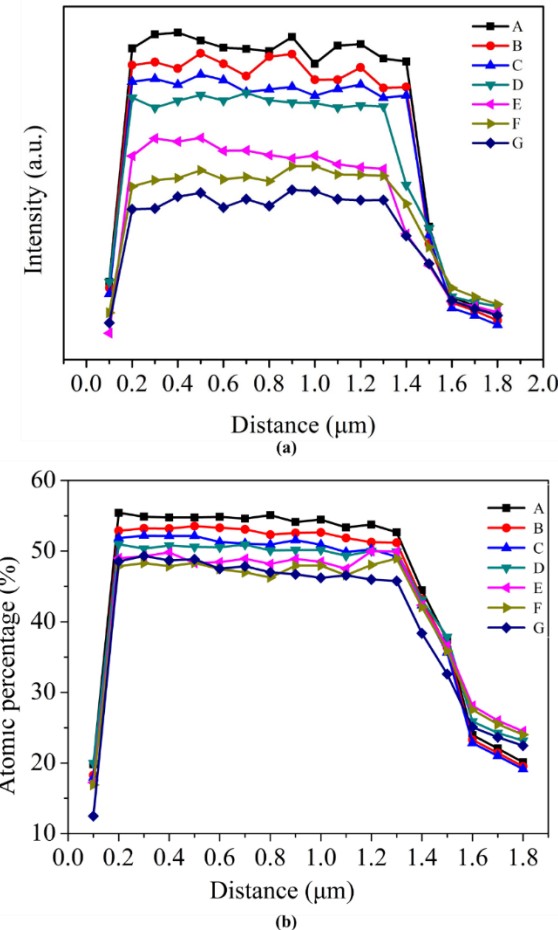

**Figure 9.** XPS-spectra of the present specimen A–G, revealing (**a**) O1*s* binding energy in each layer of Ti-6Al-4V after various polishing. (**b**) Oxygen atomic percentage in each layer of Ti-6Al-4V after various polishing. (Specimen A—electropolished for 300 s, Specimen B—electropolished for 200 s, Specimen C-electropolished for 100 s, Specimen D—chemically polished for 2 min, Specimen E—mechanically polished by #1500 SiC, Specimen F—mechanically polished by #1000 SiC, and Specimen G—obtained after CNC cutting of pristine material).

Figure 10 shows the analytical values of the bacterial culture. Figure 10a shows the data analysis of the biofilm of the bacterial culture, and the OD data is between 0.16 and 0.27. Figure 10b shows the analysis data of the number of bacterial colonies, and the CFU data are between $7 \times 10^6$ and $18 \times 10^6$.

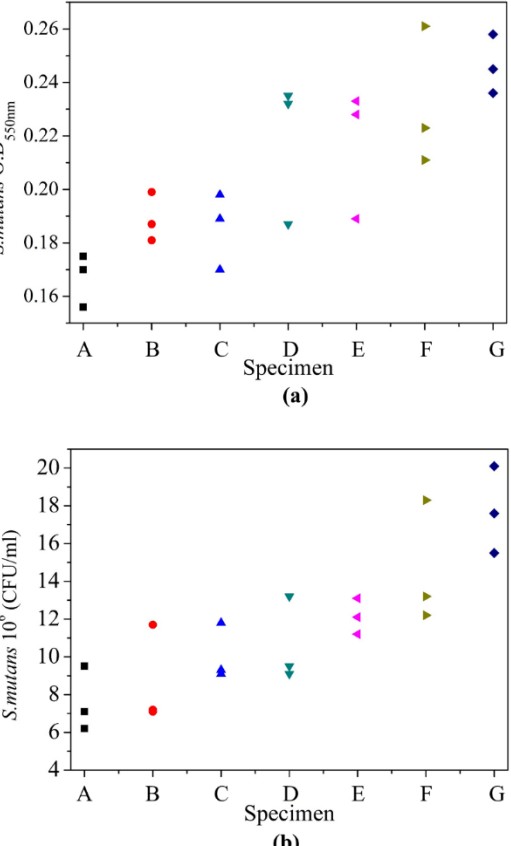

**Figure 10.** *Streptococcus mutans* bacterial amount of the present specimen A–G, revealing single-species biofilm mass and (**a**) OD$_{550\,nm}$ values and (**b**) *Streptococcus mutans* bacterial counts. (Specimen A—electropolished for 300 s, Specimen B—electropolished for 200 s, Specimen C—electropolished for 100 s, Specimen D—chemically polished for 2 min, Specimen E—mechanically polished by #1500 SiC, Specimen F—mechanically polished by #1000 SiC, and Specimen G—obtained after CNC cutting of pristine material).

Figure 11 is an SEM image of *Streptococcus mutans* on the surfaces of Ti-6Al-4V after different polishing treatments. What can be seen is a large amount of *Streptococcus mutans* covered on the surface of Ti-6Al-4V after cutting without polishing treatments (specimen G), mechanical polishing (specimen E and F) and chemical polishing (specimen D). On the contrary, the amount of *Streptococcus mutans* was decreased on the surface of Ti-6Al-4V after electropolishing (specimens A, B, and C). It is indicating that Ti-6Al-4V after electropolishing can inhibit against the biofilm formation of *Streptococcus mutans*.

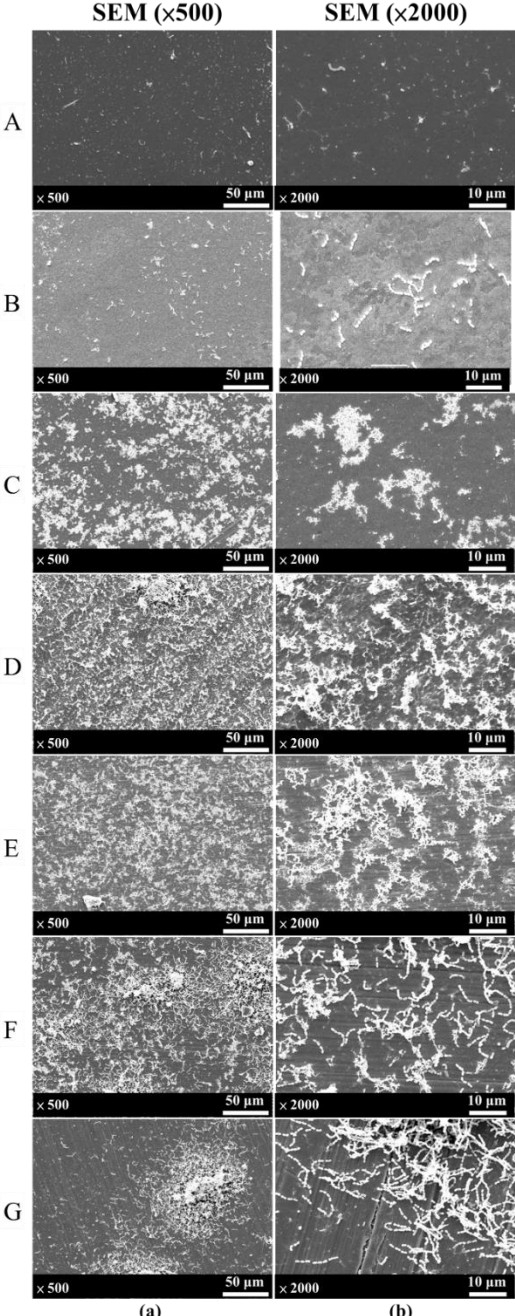

**Figure 11.** SEM micrograths taken from the Specimen A–G after biological test, showing that a large amount of *Streptococcus mutans* covered on the specimen G, specimen D, E, and F. Specimens A, B, and C are covered less amount of *Streptococcus mutans*, (**a**) Original magnification 500×, (**b**) Original magnification 2000×. (Specimen A—electropolished for 300 s, Specimen B—electropolished for 200 s, Specimen C—electropolished for 100 s, Specimen D—chemically polished for 2 min, Specimen E—mechanically polished by #1500 SiC, Specimen F—mechanically polished by #1000 SiC, and Specimen G—obtained after CNC cutting of pristine material).

Figure 12 is the scatter diagram that is drawn to represent the variability of surface roughness, contact angle, and oxygen content and its influence on the quantity of accumulated bacteria, and a linear relationship was observed.

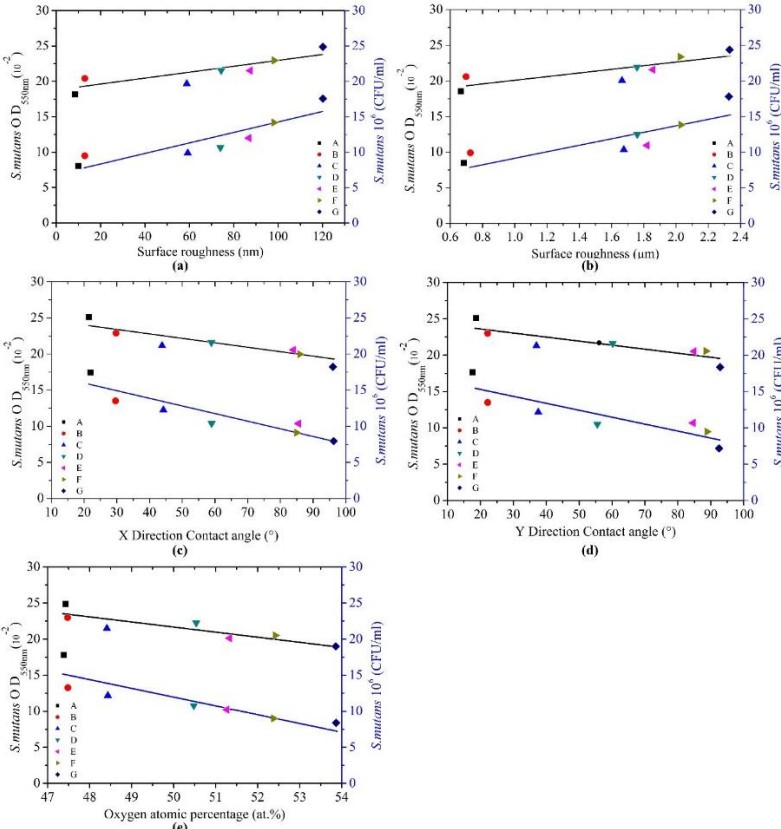

**Figure 12.** Relationship between the present of Specimen A–G surface characteristics and *Streptococcus mutans* bacterial counts (CFU/OD), It indicates that (**a**) micro/(**b**) macro surface roughness and *Streptococcus mutans* bacterial counts are positive correlation, (**c**) X direction/(**d**) Y direction contact angle and *Streptococcus mutans* bacterial counts are negative correlation, and (**e**) surface oxygen content and *Streptococcus mutans* bacterial, respectively, (Specimen A—electropolished for 300 s, Specimen B—electropolished for 200 s, Specimen C—electropolished for 100 s, Specimen D—chemically polished for 2 min, Specimen E—mechanically polished by #1500 SiC, Specimen F—mechanically polished by #1000 SiC and Specimen G—obtained after CNC cutting of pristine material).

The results show that the micro surface roughness is positively correlated with the quantity of accumulated bacteria, as shown in Figure 12a. The linear relationship equations between the OD absorbance, number of colonies, and surface roughness are $y = 0.040x + 18.80$ (the slope is 0.040) and $y = 0.074x + 6.86$ (the slope is 0.074), and there is a positive correlation between the two. The statistical analysis shows a square ratio of 0.83 and 0.83. This means that the influence between surface roughness and bacterial adhesion has an explanatory power of 83% and 83%, and $p < 0.05$ shows that the explanatory power is statistically significant.

Figure 12b shows the macro surface roughness is positively correlated with the quantity of accumulated bacteria. The linear relationship equations between the OD absorbance, number of colonies, and surface roughness is $y = 2.559x + 17.52$ (the slope is 2.559) and $y = 4.56x + 4.58$ (the slope is 4.56), and there is also a positive correlation between the two. The statistical analysis shows a square ratio of 0.72 and 0.71. This means that the influence between surface roughness and bacterial adhesion has an explanatory power of 72% and 71%.

Figure 12c shows the distribution diagram of the contact angle and the quantity of accumulated bacteria. The linear equations between the OD absorption value and the number of colonies and the surface contact angle are $y = -0.06x + 25.25$ (slope is −0.06) and $y = -0.10x + 18.09$ (slope is −0.10), respectively. This shows that the surface contact angle and the amount of bacterial adhesion are negatively correlated. Statistical analysis shows that the square ratio is 0.93 and 0.85, indicating that

the effect of the surface contact angle and the amount of bacterial adhesion has an explanatory power of 93% and 85%, respectively.

Figure 12d shows the distribution diagram of the contact angle and the quantity of accumulated bacteria. The linear equations between the OD absorption value and the number of colonies and the surface contact angle are $y = -0.06x + 24.72$ (slope is $-0.06$) and $y = -0.96x + 17.20$ (slope is $-0.96$), respectively. This shows that the surface contact angle and the amount of bacterial adhesion are negatively correlated. Statistical analysis shows that the square ratio is 0.87 and 0.80, indicating that the effect of the surface contact angle and the amount of bacterial adhesion has an explanatory power of 87% and 80%, respectively, and $p < 0.05$ shows that the explanatory power is statistically significant. The statistical analysis of the results shows that the characteristics of the surface contact angle are also important factors affecting the quantity of accumulated bacteria.

The distribution diagram of the surface oxygen content and the extent of bacterial adhesion in this study are shown in Figure 12e. The linear relationship equations between the OD absorbance value the number of colonies and the surface oxygen content are $y = -0.64x + 53.88$ (slope is $-0.64$) and $y = -1.31x + 77.65$ (slope is $-1.31$), respectively. This shows that there is a negative correlation between surface oxygen content and bacterial adhesion. The statistical analysis shows that the square ratios are 0.88 and 0.85, respectively, indicating that the effect of surface oxygen content and bacterial adhesion has an explanatory power of 88% and 85%, respectively.

In the present study one-way ANOVA statistical analysis was performed, and surface roughness, surface contact angle, and surface oxygen content had a greater impact on the amount of bacterial adhesion. The statistical results are shown in Table 3. The *F* value is the ratio of the between-group variation to the within-group variation, and the statistical representative meaning is the strength of the independent variable's influence on the dependent variable. Therefore, when the *F* value is larger and $p < 0.05$, it means that the surface characteristic has a greater influence on the quantity of accumulated bacteria. The results of this study show that the *F* values of the macroscopic surface roughness, microscopic surface roughness, surface contact angle, and surface oxygen content are 79.39, 177.23, 52.43, and 20.54, respectively. This indicates that the order of influence of the three variables of surface roughness, surface contact angle, and surface oxygen content on the adhesion of bacteria is surface roughness > surface contact angle > surface oxygen content.

**Table 3.** Results of one-way ANOVA.

| Source | Sum of Square | d*f* | Mean Square | *F* | Sig. |
|---|---|---|---|---|---|
| Macro roughness (μm) | 14.62 | 18.00 | 0.81 | 79.39 | $p < 0.05$ |
| Micro roughness (μm) | 63130.13 | 18.00 | 3507.23 | 177.23 | $p < 0.05$ |
| Contact angle (°) | 36446.45 | 18.00 | 2024.80 | 52.43 | $p < 0.05$ |
| Oxygen atomic percentage (%) | 228.29 | 18.00 | 12.68 | 20.54 | $p < 0.05$ |

d*f*: Degrees of freedom; Sig.:Significance.

## 4. Discussion

The specimens were placed in a mixture of acetic acid, perchloric acid, and glycerin for electrolytic polishing in this study. The surface reactants were quickly removed by agitating the liquid, and a smoother surface was obtained in this process. The original surface roughness of 2.34 μm/120.05 nm (specimen G: macro/micro) was reduced to 0.68 μm/10.33 nm (specimen A: macro/micro). The reduction rate was by 4–12 times. Urlea et al. [34] also used a mixture of acetic acid and perchloric acid to perform electrolytic polishing on Ti-6Al-4V, reducing the original surface roughness from approximately 3.93–22.68 to 1.28–2.52 μm, reducing it by approximately 3–9 times.

The studies have also explained the relationship between the current and voltage of Ti-6Al-4V in the electrolytic polishing process [35]. When electrolytic polishing is at a low potential of 0–14 V, a film forms on the surface of the anode that causes etching due to the passage of current. Polishing can take

place at potentials above 16 V. This study shows Ti-6Al-4V in a mixture of acetic acid and perchloric acid, with the voltage set at 25–30 V and a current of 0.5–1 A. The electrolytic polishing process can obtain a good surface roughness. This result differs from that of Urlea et al. [34] in the reduction rate of surface roughness after electrolytic polishing that is attributed to the different initial surface roughness. When the initial surface roughness value is smaller, the value obtained after electrolytic polishing is smaller [36].

In this study, the influence of surface roughness on the quantity of accumulated bacteria was understood by expressing the quantity of accumulated bacteria using the parameters of OD absorbance and the number of colonies. The results of this study reveal surface roughness significantly influence the biofilm formation and have the same trend as in some previous studies. When the surface roughness range is below 10 nm, the quantity of accumulated bacteria increases as the surface roughness increases [13,17]. When the surface roughness is between 10 and 1200 nm, it exhibits the same trend as the results of this study [37–39]. However, when the surface roughness is approximately 1860–7890 nm, Taylor et al. [39] found that although the quantity of accumulated bacteria was still greater than that of a smooth surface, there was no significant difference in the quantity of bacteria adhered to the surface. This indicates that the surface roughness of the material needs to be controlled below approximately 1800 nm to ensure antibacterial properties. In addition, some researchers also mentioned that the surface roughness is not directly related to the quantity of accumulated bacteria. The quantity of accumulated bacteria depends on the characteristics of the surface and morphology of the bacteria [40].

Directional scratches are observed on the surface after mechanical polishing. The directional scratches are adhered to easily by early colonizers for the initial step of biofilm formation, because grooves can protect bacteria to against shear forces and favor to bacterial adhesion [41]. According to Park et al., decreasing surface roughness can decrease an adhesion early-colonizer such as *S. mutans* and *S. sobrinus*, and the late-colonizer Gram-negative anaerobes such as *A. actinomycetemcomitans* and *P. gingivalis* are not significant in their effect on surface roughness, but they would be decreased with a 4-days incubation time [42]. The microstructure of Ti-6Al-4V obtained is an α + β bimodal equiaxed structure after polishing with chemical solutions, and the surface has obvious tiny protrusions on the surface. The protrusions are β island-like structures (as shown in Figure 7), and these are the surface types to which bacteria can easily adhere. However, the electropolished surface is relatively smooth, without macro scratches, and therefore it is difficult for bacteria to adhere to. The results of this study show that the surface scratches and β island structure increase the surface roughness, thereby developing the quantity of accumulated bacteria. Therefore, the surface roughness was positively correlated with the quantity of bacteria adhered. In this study, revealing that surface roughness has a more significant effect on the adhesion of early-colonizers (*S. mutans*), the late-colonizers (*A. actinomycetemcomitans and P. gingivalis*) adhere to early colonizers and do not initially adhere on tooth surfaces. Therefore, inhibiting early-colonizers may decrease late-colonizers' adhesion to prevent infections of dental implant complications.

In addition to surface roughness, some studies also mentioned that the surface contact angle affects the quantity of accumulated bacteria [13,17,18,40,43]. The present study reveals that the contact angle with X and Y direction do not show a significant difference, and thus the scratches of direction may not influence contact angle value. However, the previous study shows that anisotropic texture would affect the contact angle. Contact angle surfaces from un-textured to micro-groove textured 100–300 μm with constant depth of 10–30 μm reveal a droplet shape which becomes stretched and distorted in transformation [44]. Surface roughness transformations of the present study are low (10.33–120.05 nm and 0.68–2.34 μm), and thus anisotropic textures do not significantly influence the contact angle. Increasing contact angle can decrease bacterial adhesion. The results of this study are the same as those of some previous studies [13,17,18,43]. When the surface contact angle is between 9° and 80°, the number quantity of accumulated *S. epidermidis* [13], *E. coli* [17], and *Streptococcus* [18] bacteria decreases as the contact angle increases. This indicates that the quantity of accumulated bacteria has a negative correlation with the size of the contact angle. Additionally, the surface contact angle

must be greater than 50° in order to inhibit bacterial adhesion. Therefore, the surface may be called hydrophobic when the surface contact angle is greater than 50° [43]. The results of this study show that the contact angle needs to be greater than 88° to have good antibacterial properties. However, the results of the study by Bohinc et. al. [40] are different from the results of this study when the surface contact angle value is in a smaller range (70°–95°). There was no significant difference in the contact angle with any change in the quantity of accumulated bacteria, because bacterial adhesion depends on the different surface topography, and the adhesion of bacteria is related to the extracellular polymeric substances (EPS) produced by bacteria. However, the EPS produced by bacteria cannot adhere on the surfaces with high hydrophobicity. Therefore, the results of this study show that the surface contact angle is negatively correlated with the quantity of accumulated bacteria [17], indicating that electropolishing can obtain smooth surface and *Streptococcus mutans* are hard to attach to the surface to form a biofilm. However, a large amount of cover on the surface of Ti-6Al-4V after cutting without polishing treatments (specimen G) and mechanical polishing (specimen E and F) can be seen. *Streptococcus mutans* can adhere on the surface because of the tool mark and consistent scratches by polishing via SiC paper. The bacteria are easily crowd gathering on scratches. According to Grivet et al. [45], hydrophobic bacteria including *S. mutans*, *S. oralis*, and *S. sanguinis* showed much bacterial attachment on the hydrophobic surface. Hydrophobic surfaces are beneficial to adhere for hydrophobic bacteria. However, according to Kang et al. [8], *Streptococcus mitis* had more bacterial adhesion on the more hydrophilic surface. The present study reveals that whether surface hydrophobicity has positive or negative correlation with bacterial attachment depends on the hydrophobicity of the surface.

In addition, the surface elements and the thickness of the oxide layer also affect the adhesion of bacteria. An oxide layer on the surface of a material has the effect of inhibiting bacteria [20]. The oxide layer of the raw material (specimen G) is naturally stored in air. Because titanium alloys are highly active metals, when titanium alloys are exposed to the atmosphere, they can easily form a natural oxide layer [46]. The surface oxygen content of specimen E and F decreased after mechanical grinding. The surface of specimen D was slightly corroded that increased the activation energy of the titanium alloy and quickly combined with oxygen ions in the air after chemical polishing. Therefore, the oxide layer and oxygen content generated were more than those of the pristine titanium alloy. The pristine $TiO_2$ structure is rutile (the common crystals of titanium dioxide are rutile and anatase). Natural titanium is predominantly found as rutile titanium dioxide. However, titanium dioxide crystals with different structures can also be obtained through heat treatment and surface treatment processes [47]. The surfaces of specimens A, B, and C undergo anodic dissolution during the initial stage after electrolytic polishing; this increases the activation energy of the titanium alloy ionization process. Oxygen ions are adsorbed on the surface of the titanium alloy that diffuse and react with titanium to form a hydrophobic film of $TiO_2$. Because specimen A has a longer electropolishing time, the oxygen content is higher. The XRD results show that the electropolished $TiO_2$ structure is anatase. Chang [20] and Lin et al. [21] also found that the anatase $TiO_2$ structure facilitates good antibacterial properties of the surface of the titanium alloy. The reason for this is the release of active oxygen generated during a photocatalytic process within the $TiO_2$ nanometer-scale oxide layer. The generated $O^{2-}$, OH, and $^1O_2$ can change the permeability of the surface of the *Staphylococcus aureus* cell membrane. These free radicals then penetrate into the cell membrane to destroy the cell wall, allowing the protein or DNA to flow out of the cell membrane, causing the bacteria to lyse and die [21]. Therefore, the electrolytically polished titanium alloy can obtain better antibacterial properties than that obtained with mechanical and chemical polishing [30,48]. There may not be any direct relationship between the surface oxygen content and the quantity of accumulated bacteria. The structure of the oxide film on the surface of the titanium alloy is the main factor affecting the quantity of accumulated bacteria. Nanda et al. [49] also showed that there is no direct relationship between the thickness of the oxide film on the surface of CP-Ti and the quantity of accumulated bacteria. Therefore, the results of this study show that the linear relationship between the surface oxygen content and the amount of bacterial adhesion is negatively correlated that it is only applicable to the anatase surface oxide film of titanium alloy.

Some researchers believe that the size of the surface contact angle has a greater impact on the quantity of accumulated bacteria than the surface roughness [17]. However, Schlisselberg and Yaron [37] believe that surface roughness is the main reason for the quantity of accumulated bacteria. The results of the studies which claim that the surface features have a greater impact on the quantity of accumulated bacteria, are not consistent. Few researchers have studied the differences between surface roughness, surface contact angle, and surface oxygen content. However, according to the aforementioned discussion, it is known that the effect of the structure of the oxide film on the material surface on the antibacterial property is more important than the surface oxygen content. The surface composition of the material is not directly related to the accumulation of bacteria [37], therefore the results of this study suggest that surface roughness is the most important reason for bacterial adhesion.

## 5. Limitations

This research uses CNC cutting of the pristine titanium bar to obtain specimens for polishing. The methods include mechanical polishing, chemical polishing, and electrolytic polishing. The surface roughness value of mechanical polishing depends on the particle size of the abrasive sandpaper. The range of the surface roughness value of mechanical polishing is approximately 86–98 nm in this study. The value of surface roughness obtained via chemical polishing depends on the chemical solution and polishing time. The value of surface roughness obtained via chemical polishing is approximately 74 nm in this study. The current is concentrated on the microscopic or macroscopic rough protrusions on the surface for quick dissolution, and the melting speed is slower on the lower surface than the higher surface in the electrolytic polishing process. The electrolytic polishing process can remove the bald or burrs on the surface and make the metal surface smoother, with good glaze of the surface [50]. The value of the surface roughness is 10–58 nm. Owing to the limitations of the manufacturing process, the results of this study only apply to the influence of the surface roughness in the range of 10–100 nm on the quantity of accumulated bacteria.

## 6. Conclusions

The basic structure of Ti-6Al-4V for medical use is an equiaxed $\alpha$ phase base, with an island $\beta$ phase structure. The $TiO_2$ structure can be observed after mechanical grinding, chemical corrosion, or electrolytic polishing. The surface roughness of micro and macro are between 10 and 100 nm and 0.68 and 2.34 $\mu$m, respectively. The contact angle is between 15° and 95°. According to XPS analysis, the relative oxygen content is high within the depth of 0.2–1.4 $\mu$m of the alloy surface.

The surface characteristics of Ti-6Al-4V are a surface roughness of 10 nm, contact angle of 92°, and a relatively high oxygen content after electropolishing. It has the best bacterial inhibition. It is recommended as the best surface treatment method for Ti-6Al-4V dental implants.

The surface roughness, surface contact angle, and surface oxygen content of the material are linearly related to the quantity of accumulated bacteria after different polishing procedures. The surface characteristics of the alloy will affect the adhesion characteristics of bacteria, and the surface roughness is the most important factor affecting the amount of bacterial adhesion.

**Author Contributions:** Conceptualization, W.-C.H. and J.-K.D.; methodology, Y.-T.J. and C.-Y.C.; software, Y.-T.J.; validation, W.-C.H. and J.-K.D.; formal analysis, C.-Y.C.; investigation, Y.-T.J., W.-C.H., and J.-K.D.; resources, J.-K.D. and W.-C.H.; data curation, J.-K.D. and W.-C.H.; writing—Original draft preparation, Y.-T.J., C.-Y.C., and J.-K.D.; writing—Review and editing, J.-K.D., W.-C.H., and C.-Y.C.; visualization, W.-C.H. and J.-K.D.; supervision, C.-Y.C.; project administration, J.-K.D. and W.-C.H.; funding acquisition, J.-K.D. All authors have read and agreed to the published version of the manuscript.

**Funding:** The authors acknowledge the support from the Fund for Kaohsiung Medical University Hospital (KMUH107-7R76 and KMUH108-8M62), and the Ministry of Science and Technology (MOST 106-2314-B-037-013- and 104-2314-B-037-057-MY2).

**Conflicts of Interest:** The authors declare no conflict of interest.

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
