# Peer review of "Effects of Various Polishing Techniques on the Surface Characteristics of the Ti-6Al-4V Alloy and on Bacterial Adhesion"

_coatings, doi:10.3390/coatings10111057_

Round 1

Reviewer 1 Report

The manuscript written by Yu-Ting Jhong , Chih-Yeh Chao , Wei-Chun Hung , Je-Kang Du presents a study on the influence of different polishing techniques on characteristics of Ti-6Al-4V Alloy surfaces and bacterial adhesion conditions.

1. Before being published in Coatings journal, the authors need to address mainly the novelty of the presented study in comparison with previous ones.

The authors need also to answer to the following items:

2. The data presented in Table 1 regarding the specimens and polishing processes need to be correlated with those mentioned between lines 211-215; 232-237 (figure 1 caption), 279-280.

3. Lines 242-244 : Comment on the correspondence between alpha and beta phase of Ti-6Al-4V and the SEM images.

4. Figure 3-specify the specimen. Comment the results as function of specimens and polishing methods.

5. Figure 5a: Specify the correspondence between the represented graphs and polishing procedures and specimens.

6. Figure 5b: specify the specimen and the polishing process.

7. Comments on the influence of the polishing process on the oxygen content in 0.2-1.4 microns interval for A to G specimens.

8. Figure 4 and 6: Please specified in figure legends the meaning of A-G and their correlation with Table 1.

9. Please specify in the text and Fig 8, 9 and 10 what specimens have been used.- Please comments the results as function of different polishing techniques and specimens.

Author Response

23/10/2020

Prof. Tzer-Min Lee

Editor-in-Chief

Coatings

Dear Prof. Tzer-Min Lee:

We would like to resubmit our revised manuscript, titled " Effects of Various Polishing Techniques on the Surface Characteristics of the Ti-6Al-4V Alloy and on Bacterial Adhesion " (Manuscript ID: coatings-970781), for consideration for publication in Coatings.

   The editor and reviewers have recommended minor revisions. We have carefully reviewed these suggestions and revised our text accordingly. Please note that those words within manuscript, tables and figures are written with red words for correction. Our responses to each of the reviewers’ comments are detailed below. The details are provided in the manuscript and mark with red for correction.

Comments and Suggestions for Authors:…………

Reply to the Comments and Suggestions for Reviewer 3

The manuscript written by Yu-Ting Jhong , Chih-Yeh Chao , Wei-Chun Hung , Je-Kang Du presents a study on the influence of different polishing techniques on characteristics of Ti-6Al-4V Alloy surfaces and bacterial adhesion conditions.

1. Before being published in Coatings journal, the authors need to address mainly the novelty of the presented study in comparison with previous ones.

We appreciate this helpful suggestion from the reviewer. The introduction had been revised, it describes the novelty of the presented study. In addition, seldom information of surface roughness, contact angle, and surface oxygen content on the quantity of accumulated bacteria are not consistent at the same time. The present study would like to research these three factors influence of the aforementioned three factors on the quantity of accumulated bacteria. This information has been supplemented at line 37-105 Page 1~3 of revised manuscript and marked with red.

The authors need also to answer to the following items:

2. The data presented in Table 1 regarding the specimens and polishing processes need to be correlated with those mentioned between lines 211-215; 232-237 (figure 1 caption), 279-280.

Thank you for your deep concern. We had revised the Table 1, of which described the definition of specimen A to G at Page 3 of revised manuscript. The data and results are described in Table 2. and Figure 3. These informations have been supplemented in the section of materials and results at Page 3,5,7 of revised manuscript.

3. Lines 242-244 : Comment on the correspondence between alpha and beta phase of Ti-6Al-4V and the SEM images.

Thank you for reviewer’s comments. we added correspondence between alpha and beta phase of Ti-6Al-4V and the SEM images showed in Fig 7. This information has been supplemented at line 282-287 Page 11~12 of revised manuscript and marked with red.

4.Figure 3-specify the specimen. Comment the results as function of specimens and polishing methods.

Thank you for your pointing out. The Figure 3 had changed to Figure 5 and we added function of specimens and polishing methods in Figure 5 at Page 10 of revised manuscript.

5.Figure 5a: Specify the correspondence between the represented graphs and polishing procedures and specimens.

We appreciate this helpful comment from the reviewer. The Figure 5a had changed to Figure 8a. We revised figure’s explanation and identified the specimens of legend at line 306-312 Page 13 of revised manuscript and marked with red.

6.Figure 5b: specify the specimen and the polishing process.

Thank you for your pointing out. The Figure 5b had changed to Figure 8b. We revised figure’s explanation and identified the specimens of legend at line 306-312 Page 13 of revised manuscript and marked with red.

7.Comments on the influence of the polishing process on the oxygen content in 0.2-1.4 microns interval for A to G specimens.

Thank you for your kind suggestion. The results are revised, and the order of oxygen content for various polishing process is electropolishing > chemical polishing > mechanical polishing. This information has been supplemented at line 321-326 Page 14 of revised manuscript and marked with red.

8.Figure 4 and 6: Please specified in figure legends the meaning of A-G and their correlation with Table 1.

Thank you for your question to make our data clear. Figure 4 and 6 are changed to Figure 7 and 9 and the figure legends have been revised at line 289-295 and 328-331 Page 12,14 of revised manuscript and marked with red. The meaning of specimen A-G are showed in Table 1 at Page 3 of revised manuscript .

9.Please specify in the text and Fig 8, 9 and 10 what specimens have been used.- Please comments the results as function of different polishing techniques and specimens.

Thank you for your kind suggestion. We had combined Figures 8, 9, 10 into Fig 12 at Page 18 of revised manuscript. The legends of specimen A-G are added and revised in the text. We discuss the results as function of different polishing techniques and specimens at line 358-396 Page 17 of revised manuscript and marked with red.

Thank you for your consideration. We hope our manuscript is suitable for publication in your journal.

Sincerely,

Je-Kang Du

Associate professor, School of Dentistry, Kaohsiung Medical University

100 Shih-Chuan 1st Road, San-Ming District, Kaohsiung, Taiwan 807

Tel.: + 886-7-3121101 ext. 7006, Fax: + 886-7-3121510

Reviewer 2 Report

The theme is interesting and current. Bibliographic support is adequate. However, there are some flaws in the study design.

1.The introduction is well written, although very long. Authors should summarize, focusing more on the topic.

2.The point 2 should be materials and methods.

3. In point 2.1, the authors do not describe the positive and negative control groups. How were they carried out? The results of these groups must be shown in the graphical representation so that a comparison can be made between the control and experimental groups. If the authors did not carry out control groups they must redo the study design and include them.

4. Table 1 represents results. Its position in the article should be reformulated. Must be placed in the results section.

5. The results should be explained in more detail, demonstrating the differences between the different groups.

6. The graphs of the discussion must be included in the results chapter. Authors should redo the entire discussion that should compare their results with other studies, only.

Author Response

23/10/2020

Prof. Tzer-Min Lee

Editor-in-Chief

Coatings

Dear Prof. Tzer-Min Lee:

We would like to resubmit our revised manuscript, titled " Effects of Various Polishing Techniques on the Surface Characteristics of the Ti-6Al-4V Alloy and on Bacterial Adhesion " (Manuscript ID: coatings-970781), for consideration for publication in Coatings.

   The editor and reviewers have recommended minor revisions. We have carefully reviewed these suggestions and revised our text accordingly. Please note that those words within manuscript, tables and figures are written with red words for correction. Our responses to each of the reviewers’ comments are detailed below. The details are provided in the manuscript and mark with red for correction.

Comments and Suggestions for Authors:…………

Reply to the Comments and Suggestions for Reviewer 2

The theme is interesting and current. Bibliographic support is adequate. However, there are some flaws in the study design.

  1. The introduction is well written, although very long. Authors should summarize, focusing more on the topic.

Thank you for reviewer’s suggestions. The description of introduction has been revised.

  1. The point 2 should be materials and methods.

Thank you for your pointing out. We have corrected the point 2 into materials and methods.

  1. In point 2.1, the authors do not describe the positive and negative control groups. How were they carried out? The results of these groups must be shown in the graphical representation so that a comparison can be made between the control and experimental groups. If the authors did not carry out control groups they must redo the study design and include them.

Thank you for your question. The control and experimental groups are shown in Table 1. at Page 3 of revised manuscript. The results of these groups showed in Table 2. at Page 5 of revised manuscript.

  1. Table 1 represents results. Its position in the article should be reformulated. Must be placed in the results section.

Thank you for your deep concern. We had revised the Table 1, of which described the definition of specimen A to G at Page 3 of revised manuscript. The data and results are described in Table 2 and placed in the section of results at Page 5 of revised manuscript.

  1. The results should be explained in more detail, demonstrating the differences between the different groups.

Thank you for reviewer’s suggestions. We had added some explanations (including Figures and Tables) in the section of Results. This information has been supplemented at line 196-418 Page 5~19 of revised manuscript and marked with red.

  1. The graphs of the discussion must be included in the results chapter. Authors should redo the entire discussion that should compare their results with other studies, only.

All graphs of the discussion are revised and move to the section of Results. The entire section of discussion have been revised and we had compared the results with other studies. This information has been supplemented at line 419-543 Page 19~21 of revised manuscript and marked with red.

Thank you for your consideration. We hope our manuscript is suitable for publication in your journal.

Sincerely,

Je-Kang Du

Associate professor, School of Dentistry, Kaohsiung Medical University

100 Shih-Chuan 1st Road, San-Ming District, Kaohsiung, Taiwan 807

Tel.: + 886-7-3121101 ext. 7006, Fax: + 886-7-3121510

Reviewer 3 Report

In this study the authors have shown that the surface roughness is positively correlated with the quantity of accumulated bacteria, the surface contact angle and the amount of bacterial adhesion are negatively correlated and there is a negative correlation between surface oxygen content and bacterial adhesion.

1. After statistical analyses, the author conclude that the order of influence of the three variables on the adhesion of bacteria is surface roughness > surface contact angle > surface oxygen content. The authors also claim that their study showed that electrolytic polishing provides the best surface treatment of Ti-6Al-4V in dental implants. The authors go too far on this point because they only study the adhesion of a single type of bacteria (Streptococcus mutans). The authors have chosen to study Streptococcus mutans, however it is not the bacteria mainly responsible for peri-implantitis. A choice like Porphyromonas gingivalis would have been more relevant. They must justify their choice in the introduction or talk about it in the discussion. The manuscript requires major revision on M&M and figures. Authors must absolutely clarify the name of the samples! The names of samples are sometimes reversed in the figures and in the results.

I have attempted to describe clarifications which I feel are necessary below:

2.Figure 1 : Unreadable axis units. Error in the legend on the names of the samples (they are reversed). There is the same error line 212-214 in the results (names are reversed from Table 1).

3.Clearly there is a problem with specimen identification in the different figures! For example, Specimen F is “red” in figure 4b whereas the same specimen F is “green on figure 7. All the colors are reversed between figure 4 and 7 ! Please clarify this point.

4.Line 165 : Since, a mechanical polishing generates a type of anisotropic texture, the profile of the droplet tends to be different when viewed at different directions. What was the direction (parallel or perpendicular) of the acquired image of the droplet profile on the mechanical polished surface regarding to the polish direction? Did the authors have measurements in both direction? Please, provide the proper explanation in the text;

5.Figures 8, 9, 10 should be combined into one figure. Each point should be identified to know which sample it is.

6.In Figure 8, the authors analyze the influence of micro-roughness but what about the influence of macro-roughness?

7.SEM images of Strepotococcus mutans on the different surfaces must be added to the results to observe the interaction between bacteria and surface roughness.

8.Part of the discussion actually corresponds to a description of results, there are moreover figures of results inserted in the discussion. Author have to put them in the right part.

9.In the discussion about surface roughness and bacterial adhesion, it should be added that the morphology of the topography is very important, for example we find in nature surface morphologies with nanospikes which are bactericidal (for example the wings of cicadas).

10.Discussion: It is shown by the literature that bacterial adhesion depends on whether they are GRAM + or GRAM-. Streptococcus is GRAM + but other bacteria that cause peri-implantitis like P. Gingivalis are GRAM-. Would they have the same adhesion? This point must be discussed.

Author Response

23/10/2020

Prof. Tzer-Min Lee

Editor-in-Chief

Coatings

Dear Prof. Tzer-Min Lee:

We would like to resubmit our revised manuscript, titled " Effects of Various Polishing Techniques on the Surface Characteristics of the Ti-6Al-4V Alloy and on Bacterial Adhesion " (Manuscript ID: coatings-970781), for consideration for publication in Coatings.

   The editor and reviewers have recommended minor revisions. We have carefully reviewed these suggestions and revised our text accordingly. Please note that those words within manuscript, tables and figures are written with red words for correction. Our responses to each of the reviewers’ comments are detailed below. The details are provided in the manuscript and mark with red for correction.

Comments and Suggestions for Authors:…………

Reviewer1 :

In this study the authors have shown that the surface roughness is positively correlated with the quantity of accumulated bacteria, the surface contact angle and the amount of bacterial adhesion are negatively correlated and there is a negative correlation between surface oxygen content and bacterial adhesion.

1.After statistical analyses, the author conclude that the order of influence of the three variables on the adhesion of bacteria is surface roughness > surface contact angle > surface oxygen content. The authors also claim that their study showed that electrolytic polishing provides the best surface treatment of Ti-6Al-4V in dental implants. The authors go too far on this point because they only study the adhesion of a single type of bacteria (Streptococcus mutans). The authors have chosen to study Streptococcus mutans, however it is not the bacteria mainly responsible for peri-implantitis. A choice like Porphyromonas gingivalis would have been more relevant. They must justify their choice in the introduction or talk about it in the discussion. The manuscript requires major revision on M&M and figures. Authors must absolutely clarify the name of the samples! The names of samples are sometimes reversed in the figures and in the results.

Thank you for comments of Reviewer 1. The manuscript has revised. The strain of Streptococcus was called “early colonizers” because they take part in formation of early attachment of biofilm. Strain of Streptococcus can make a lot of extracellular polysaccharides when they gain the sucrose. And then, they can strengthen mechanical property and adhesiveness of biofilm and cause biological complications. This study is revealing that surface roughness have more significant effect on the adhesion of early-colonizers (S. mutans). Previous study is showing the late-colonizers (A. actinomycetemcomitans and P. gingivalis) adhere to early colonizers and don’t initially adhere on tooth surfaces. Therefore, inhibiting early-colonizers may decrease late-colonizers adhesion to prevent infections of dental implant complications. Therefore, it is necessary to decrease early bacterial attachment to prevent biological complications.

I have attempted to describe clarifications which I feel are necessary below:

2.Figure 1 : Unreadable axis units. Error in the legend on the names of the samples (they are reversed). There is the same error line 212-214 in the results (names are reversed from Table 1).

 Thank you for the reviewer’s suggestions. The errors had been corrected in the revised manuscript and marked with red.

3.Clearly there is a problem with specimen identification in the different figures! For example, Specimen F is “red” in figure 4b whereas the same specimen F is “green on figure 7. All the colors are reversed between figure 4 and 7 ! Please clarify this point.

Thank you for your pointing out. We had corrected the errors in the revised manuscript and marked with red.

4.Line 165 :Since, a mechanical polishing generates a type of anisotropic texture, the profile of the droplet tends to be different when viewed at different directions. What was the direction (parallel or perpendicular) of the acquired image of the droplet profile on the mechanical polished surface regarding to the polish direction? Did the authors have measurements in both direction? Please, provide the proper explanation in the text;

We appreciate this helpful comment from the reviewer. The manuscript also revised. This information has been supplemented at line 152-154 and Fig 1. Page 4 of revised manuscript and marked with red. In addition, it is revealing that the contact angle between parallel and perpendicular direction is significant difference. The variant of contact angle in parallel and perpendicular are in the range of 0.86°-7.74°, and the variant of micro and macro surface roughness are 10.33-120.05 nm, and 0.68-2.34μm, respectively.

5.Figures 8, 9, 10 should be combined into one figure. Each point should be identified to know which sample it is.

Thank you for your kind suggestion. We had combined Figures 8, 9, 10 into Fig 12 at Page 18 of revised manuscript.

6.In Figure 8, the authors analyze the influence of micro-roughness but what about the influence of macro-roughness?

Thank you for your question to make our data clear. The influence of macro-roughness has added in figure 12(b) at Page 18 of revised manuscript. The macro surface roughness is positively correlated with the quantity of accumulated bacteria. The linear relationship equations between the OD absorbance, number of colonies, and surface roughness is y=2.559x+17.52 (the slope is 2.559) and y=4.56x+4.58 (the slope is 4.56), and there is also a positive correlation between the two. The statistical analysis shows a square ratio of 0.72 and 0.71. This means that the influence between surface roughness and bacterial adhesion has an explanatory power of 72% and 71%. This information has been supplemented at line 368-373 Page 17 of revised manuscript and marked with red.

7.SEM images of Strepotococcus mutans on the different surfaces must be added to the results to observe the interaction between bacteria and surface roughness.

Thank you for your suggestions. The SEM images of Strepotococcus mutans are added in Figure 11. at Page 16 of revised manuscript. It can be seen a large amount of Streptococcus mutans covered on the surface of Ti-6Al-4V after cutting without polishing treatments (specimen G), mechanical polishing (specimen E and F) and chemical polishing (specimen D). On the contrary, amount of Streptococcus mutans was decrease on the surface of Ti-6Al-4V after electropolishing (specimens A, B and C). It is indicating that Ti-6Al-4V after electropolishing can inhibit against the biofilm formation of Streptococcus mutans. This information has been supplemented at line 345-349 Page 15 of revised manuscript and marked with red.

8.Part of the discussion actually corresponds to a description of results, there are moreover figures of results inserted in the discussion. Authors have to put them in the right part.

Thank you for reviewer’s suggestions. All figures are putted and described in the section of results.

9.In the discussion about surface roughness and bacterial adhesion, it should be added that the morphology of the topography is very important, for example we find in nature surface morphologies with nanospikes which are bactericidal (for example the wings of cicadas).

Thank you for your question. The directional scratches are adhered easily by early colonizers for initial step of biofilm formation, because grooves can protect bacteria to against shear forces and favor to bacterial adhesion. The microstructure of Ti-6Al-4V obtained is an α+β bimodal equiaxed structure. The protrusions are β island lead bacteria easily to adhere. However, the electropolished surface is relatively smooth, without macro scratches, and therefore difficult for bacteria to adhere. This information has been supplemented at line 450-467 Page 19~20 of revised manuscript and marked with red.

10.Discussion: It is shown by the literature that bacterial adhesion depends on whether they are GRAM + or GRAM-. Streptococcus is GRAM + but other bacteria that cause peri-implantitis like P. Gingivali are GRAM-. Would they have the same adhesion? This point must be discussed.

 We appreciate this helpful comment from the reviewer. Decreasing surface roughness can decrease adhesion S. mutans adhesion. The late-colonizer gram-negative anaerobes such as A. actinomycetemcomitans and P. gingivalis are not significant effect on surface roughness, but they would be decreased with a 4-days incubation time. This information has been supplemented at line 450-467 Page 19~20 of revised manuscript and marked with red.

Thank you for your consideration. We hope our manuscript is suitable for publication in your journal.

Sincerely,

Je-Kang Du

Associate professor, School of Dentistry, Kaohsiung Medical University

100 Shih-Chuan 1st Road, San-Ming District, Kaohsiung, Taiwan 807

Tel.: + 886-7-3121101 ext. 7006, Fax: + 886-7-3121510

Round 2

Reviewer 1 Report

I recomand the publication of the manuscript in current form.

Please check the english language and style.

Reviewer 2 Report

The changes made by the authors substantially improved the article, especially in the sub-chapters of materials and methods and results.
I accept the article for publication in present form.